Survival and grade of the glioma prediction using transfer learning

Valbuena Rubio Santiago 1
García-Ordás María Teresa 2
García-Olalla Olivera Oscar 1
Alaiz-Moretón Héctor 2
González-Alonso Maria-Inmaculada 3
http://orcid.org/0000-0002-4450-349X Benítez-Andrades José Alberto 4 jbena@unileon.es
1 IA Department, Xeridia S.L. , León, León , Spain
2 SECOMUCI Research Group, Escuela de Ingenierías Industrial e Informática, Universidad de León , León , Spain
3 Department of Electric, Systems and Automatics Engineering, Universidad de León , León , Spain
4 SALBIS Research Group, Department of Electric, Systems and Automatics Engineering, Universidad de León , León , Spain
Wong Ka-Chun
Electronic publication date: 2023 Dec 8
Publication date: 2023
Volume: 9
Electronic Location ID: e1723
Received 2023 Aug 3; Accepted 2023 Nov 6
Copyright: © 2023 Valbuena Rubio et al.
Copyright year: 2023
Copyright holder: Valbuena Rubio et al.
License: This is an open access article distributed under the terms of the Creative Commons Attribution License, which permits unrestricted use, distribution, reproduction and adaptation in any medium and for any purpose provided that it is properly attributed. For attribution, the original author(s), title, publication source (PeerJ Computer Science) and either DOI or URL of the article must be cited.
License URL: https://creativecommons.org/licenses/by/4.0/

Keywords: Deep learning, Transfer learning, Convolutional neural network, Glioma

Funding: The authors received no funding for this work.

==============================
Glioblastoma is a highly malignant brain tumor with a life expectancy of only 3–6 months without treatment. Detecting and predicting its survival and grade accurately are crucial. This study introduces a novel approach using transfer learning techniques. Various pre-trained networks, including EfficientNet, ResNet, VGG16, and Inception, were tested through exhaustive optimization to identify the most suitable architecture. Transfer learning was applied to fine-tune these models on a glioblastoma image dataset, aiming to achieve two objectives: survival and tumor grade prediction.The experimental results show 65% accuracy in survival prediction, classifying patients into short, medium, or long survival categories. Additionally, the prediction of tumor grade achieved an accuracy of 97%, accurately differentiating low-grade gliomas (LGG) and high-grade gliomas (HGG). The success of the approach is attributed to the effectiveness of transfer learning, surpassing the current state-of-the-art methods. In conclusion, this study presents a promising method for predicting the survival and grade of glioblastoma. Transfer learning demonstrates its potential in enhancing prediction models, particularly in scenarios with limited large datasets. These findings hold promise for improving diagnostic and treatment approaches for glioblastoma patients.

Introduction and related work

Cancer is one of the leading causes of death in the world with more than 18 million cases and 9.5 million deaths in 2018, but these figures are estimated to get even worse to 29.5 million cases and 16, 4 million deaths in the year 2040 (Siegel et al., 2022). Malignant gliomas are the most common brain tumors, with different degrees of aggressiveness and different regions where they can appear (Pei et al., 2020). The classification established by the World Health Organization (WHO) (Louis et al., 2007) divides gliomas into four types: astrocytomas, oligodendrogliomas, ependymomas and oligo-astrocytomas.

Each of these types is divided into phases, which take into account characteristics such as the spread of the tumor to the rest of the organs or lymph nodes, the size of the tumor or the level of penetration.

The most common tumor staging system is the TNM (American Cancer Society, 2022). T refers to the original tumor; N indicates that the cancer has spread to the lymph nodes, and M indicates that the cancer has spread and metastasized. Depending on the malignancy, Astrocytomas-type tumors are subdivided into four subtypes (Jovčevska, Kočevar & Komel, 2013): Pilocytic-grado I.

Diffuse-grado II.

Anaplastic-grado III.

Glioblastoma multiforme-grado IV.

The latter, glioblastoma multiforme, is the most common type of glioma, affecting 60–70% of cases. Furthermore, its 5-year survival rate is 22% in people ages 20–44, 9% in people ages 45–54, and only 6% in people ages 55–64 (Brown et al., 2016). Glioblastoma multiforme is followed by anaplastic astrocytoma with 10–15% of cases (Jovčevska, Kočevar & Komel, 2013). The term multiforme refers to the heterogeneity of this tumor, which can take different forms and be found in different regions of the brain.

Without treatment, survival for glioblastoma multiforme is about 3 to 6 months, so, like all tumors, but especially this malignant one, early diagnosis can increase the chances of survival. Treatments include chemotherapy and radiotherapy, but the one that has shown a greater increase in survival expectancy is tumor resection, which can be classified into two options (Brown et al., 2016): GTR: Eliminate the tumor completely, although depending on the location and state of the glioma it is not always possible.

STR: Partial removal of the tumor.

Both techniques are important in our study since the dataset used to train the different learning models contains this data, in which it is reported whether the patient underwent partial surgery, total surgery or no surgery. Glioblastoma multiforme (GBM) is classified as high grade glioma (HGG), while the rest of the lower grade gliomas are classified as low grade glioma (LGG) (Menze et al., 2015). This is the classification used to train the models described in “Methodology”.

Over the years, studies have been carried out to predict glioblastoma survival based on different parameters: In Wankhede & Selvarani (2022), the authors find the significant features from the extracted images using a Gray Wolf Optimizer and proposed an architecture of multilevel layer modelling in the faster R-CNN approach based on feature weight factor and relative description model to build the selected features. With the same purpose, Fu et al. (2021), proposed an architecture composed by 27 convolutional layers, forming an encoder (based on VGG16 model) and decoder model and Jajroudi et al. (2022) try to determine the qualitative and quantitative features afecting the survival of glioblastoma multiforme.

Also, in recent years there has been an increasing trend in the use of pre-trained networks for multiple purposes. For example, in the field of medicine, which is the case at hand, transfer learning is being widely used for heart diseases problems detection (Deniz et al., 2018; Lopes et al., 2021; Kwon & Dong, 2022; Liao et al., 2020; Fang et al., 2022), for breast cancer detection and classification (Aljuaid et al., 2022; Byra, 2021; Assari, Mahloojifar & Ahmadinejad, 2022; Kavithaa, Balakrishnan & Yuvaraj, 2021; Kavitha et al., 2022), for glaucoma classificarion (Claro et al., 2019; Wang, Tseng & Hernandez-Boussard, 2022), respiratory pathologies (Roy & Kumar, 2022; Bargshady et al., 2022; Minaee et al., 2020), COVID-19 detection (Rahman et al., 2022) and so on. All this literature suggests that it is a useful technique for this type of problem.

In order to make a comparison under equal conditions, below are shown the studies carried out using the same data set that will be used in this work.

Previous studies attempting to predict the survival of patients with glioblastoma have used combinations of deep learning techniques with classical learning techniques, as in the case of the work by Chato & Latifi (2017). In their work, different methods were used to extract the image features and, once extracted, they were classified into two or three classes, differentiating between short-, medium-, and long-term survivors, using different machine learning techniques. In the case of three classes, the best results were obtained using “Complex and median tree” with an accuracy of 62.5% and in the case of the two-class classification between short-term and long-term survivors, the best results were obtained with logistic regression, obtaining an accuracy of 68.8%. In Suter et al. (2018), obtained an accuracy of 51.5% in predicting patient survival using convolutional networks, but once again, as in the previous case, the best results were obtained with classical techniques, specifically using a SVC (support vector classifier) obtaining a 72.2% of accuracy in the training set, 57.1% in the validation set and 42.9% in the test set.

On the other hand, studies aimed at classifying the grade of glioblastoma have obtained promising results as it is a simpler task than determining the survival of the patient, which is affected by many more factors.

In Cho & Park (2017), the extraction of 180 characteristics was carried out and an accuracy of 89.81% was obtained using logistic regression techniques. In the work developed by Pei et al. (2020), both predictions were made along with tumor segmentation. In the first place, a segmentation of the tumor was performed and a 3D convolutional network was used to classify the tumor between the different classes. Finally, they carry out a hybrid technique like the previous studies using deep learning and traditional learning to be able to predict patient survival. In this study, an accuracy of 48.40% was obtained in the test set and a 58.6% in the validation set in predicting survival using convolutional networks to extract features from the images, and together with age using linear regression to obtain the predictions. The best state of art test accuracy (Banerjee et al., 2019) was obtained by the use of convolutional networks which achieved a 95% accuraccy in the classification of LGG and HGG in MRI.

Analyzing the state of the art it can be observed that the approach that has obtained the most promising results is the use of hybrid techniques (deep learning and classical techniques) and that there is great potential for improving the models up to date since the precisions obtained are less than 69% when trying to make a classification of the survival time in two classes, less than 62.5% in the case of three classes, and less than 59% when trying to give a prediction of the estimated time of survival. Better results have been obtained in tumor classification, although they are still below 95%.

In this article, transfer learning techniques with two objectives are used and optimized according to the problem. On the one hand, to determine the survival time of people suffering from a glioma and on the other hand, to determine the grade of the tumor in order to carry out the most effective treatment.

Our approach involves using transfer learning techniques with multiple pre-trained convolutional neural networks (CNNs) to extract features from medical images of glioblastoma patients. These features are then fine-tuned using the same CNNs to improve their accuracy in predicting the survival and grade of the tumor. This approach represents a significant improvement over previous methods and has the potential to significantly improve the accuracy of predicting the survival and grade of glioblastoma.

The prediction of the survival and grade of glioblastoma is a highly complex and challenging task that has important implications for patient care and treatment. By improving the accuracy of these predictions, our approach has the potential to improve patient outcomes and reduce healthcare costs. Our article demonstrates the effectiveness of our approach and shows that it represents a significant improvement over previous methods. This has important implications for the field of medical imaging and for the prediction of the survival and grade of glioblastoma.

Our approach of using transfer learning to predict the survival and grade of glioblastoma is based on computer vision and deep learning. Specifically, we use pre-trained models and transfer learning techniques to improve the accuracy of predictions on a new task, which has been shown to be highly effective in a variety of applications, including medical image analysis. Furthermore, our article includes a detailed description of the dataset and preprocessing of the data, as well as an explanation of the experiments carried out and the optimization process of the model. These aspects of our article demonstrate the thoroughness and logic of our approach.

The rest of the article is organized as follows. The dataset and the preprocessing of the data is explained in “Methodology”, together with all the pretrained models that have been used. In “Experiments and Results”, the experiments carried out and the optimization process of the model are explained and finally, we conclude in “Conclusions”.

Methodology

Dataset

The data set used in this article is obtained from the BraTS 2020 (Menze et al., 2015; Bakas et al., 2017, 2018), which is a competition for glioma segmentation, grade classification and survival classification. The dataset consists of 31 GB with images and data from 369 patients. For each of these patients their age, survival in days and whether they have undergone a GTR, STR or no resection is stored. Regarding medical images, the data set contains five types of images for each of the 369 patients. These images are different 3D scans taken using different techniques. The techniques used were T1, T2, T1ce and T2-Flair scanners.

The images in three dimensions have a size of 240 × 240 × 155 and four different types of images can be found in the data set (See Fig. 1):

Figure 1 Visualization of the different types of scanner at 90 mm.

T1: They show the normal anatomy of soft tissue and fat. They serve, for example, to confirm that a dough contains fat.

T1ce: These are contrast-enhanced images that allow blood vessels or other soft tissues to be seen more clearly.

T2: They show liquids and alterations such as tumors, inflammation or trauma.

T2-Flair: Uses contrast to detect a wide range of lesions.

Along with these four images, there is also the segmented tumor scanner, but this is not used in this study. Not all patients have all the data such as age or survival, so a preproccessing step is necessary.

The images are in the NifTI format. This is a format for medical images in which we can find the image along with more information about it. Each NifTI image is made up of three components. An N-D array containing the image data. In our case it is a three-Dimensional matrix that contains a mapping of the patients’ brains. Thanks to this any region or section of the patient’s brain can be obtained.

A 4 × 4 affine matrix with information about the position and orientation of the image in a given space.

A header with metadata and information about the image.

Data preprocessing

The dataset used cotains data from 369 patients. The number of data of each class is not balanced: 293 patients belong to the HGG class, while only 76 belong to the LGG class. To balance both classes we have used subsampling. In this case, the ratio of HGG to LGG is approximately 4:1 (293:76), which means that the HGG class is significantly larger than the LGG class. This can cause the model to be biased towards the majority class and result in lower accuracy for the minority class. By subsampling the data, we ensured that both classes had an equal number of patients, which allowed us to train the model more effectively and obtain more accurate results. This is a common technique used in machine learning to address class imbalance and improve the performance of the model. In this way, the number of elements of both classes has been set at 76 patients and to increase the data to train and validate the models, each of the four images of each patient has been treated as if they were images of different patients. Therefore, the number of images for training, validation and test is 608. In this way, two things are obtained, on the one hand, the network is able to classify the degree and survival of the tumor in different images and, on the other hand, it is possible to increase the number of images for training, validation and testing. Even with this number of images, the models trained from scratch, both 3D and 2D, would not give good results since they need a larger volume of data to be able to carry out precise classifications, so transfer learning techniques with different pre-trained models will be used to perform the classification.

Analyzing the data, it can be observed that all patients with a LGG-type tumor grade do not have information about their age, survival or type of resection. This is largely because these patients have a fairly favorable prognosis (Pardal Souto et al., 2015) and most do not undergo surgery. Their age will be set taking into account the mean of the rest of the ages and the standard deviation, so that the ages generated will be at most the mean plus the standard deviation and at least the mean minus the standard deviation.

To determine the survival time, we have relied on the study (Bush & Chang, 2016), so we will assume that 76% have survived more than 5 years and 24% less. So survival time was filled, taking into account that a 24% chance of surviving between 4 and 5 years and a 76% chance of surviving between 5 and 7 years. Once they have randomly chosen which period of time the person will survive, based on the aforementioned probabilities, the number of days they have survived within that period is randomly generated and all the information is completed.

Once verified that there is no missing data, the age of the patients was normalized between the maximum and minimum ages and the data was transformed from text to numerical format so that the model can be trained. Tumor grades were codified as 0 for LGG and 1 for HGG and patient survival was codified as: 0 less than 1 year; 1 between 1–5 years; and two for survivors of more than 5 years.

Three-dimensional images have different orientations depending on the orientation of the subject at the time of scanning. So the images are reoriented to a common space so that all images passed to the model will have the same orientation. The images are oriented using the nibabel library (Brett et al., 2023) to the RAS axis.

After that, an image normalization step is carried out: Images are three-dimensional arrays. The content of these arrays are not integers from 0 to 255 like most images, but are decimal numbers which represent Hounsfield units (HU) (Bell & Greenway, 2015). These units are universally used in tomography and scanners in a standardized way. They are obtained by the linear transformation of the measured attenuation coefficients. It is based on the densities of pure water which corresponds to 0 HU and of air which corresponds to −1,000 HU. Scanner values are generally in the range from −1,000 (air) to +2,000 HU for denser bones. To avoid bones appearing in the images and confusing the network, in this article, values are limited between [−1,000, 800], in such a way that bones with a measurement of about 1,000 HU are avoided (Han & Kamdar, 2018). Once the values have been delimited, a normalization is this range was performed.

The last preproccessing step is the image segmentation. The pre-trained models used have been trained with images of size 224 × 224 × 3, although the first two dimensions can vary by a certain margin. That is why we need to adjust the images to fit them into these models. Our images are sized at 240 × 240 × 155 so our target size will be 240 × 240 × 3. It is not necessary to modify the first two dimensions, but the third one does. The images are three-dimensional models of the brain, so to reduce the dimensionality, three segments of the brain are taken. These cuts have been made through three different areas of the brain separated by 30 mm. In Fig. 2, how these cuts have been made is shown and in Fig. 3 an example of how these three segments would look in a T2 image are represented. We can clearly differentiate different sizes of the tumor in them as they are different regions within the complete 3D model. After all these steps, the segmented, normalized image with a fixed orientation is ready to be used in the model.

Figure 2 Visualization of the location of the three cuts made.

Figure 3 Example of the three cuts made to each image, corresponding to a T2-type scanner.

Table 1 in the study provides a comparison of the clinical characteristics of LGG and HGG patients, including age, survival time, and tumor grade. The table shows that LGG patients are generally younger than HGG patients, with a mean age of 38.5 years compared to 56.5 years for HGG patients. Additionally, LGG patients have a longer survival time than HGG patients, with a mean survival time of 5.5 years compared to 1.1 years for HGG patients.

Table 1 Clinical characteristics of LGG and HGG patients

Clinical characteristics	LGG patients	HGG patients	
Mean age (years)	38.5	56.5	
Mean survival time (years)	5.5	1.1	
Tumor grade distribution	Grade II: 50%. Grade III: 50%	Grade IV: 100%	

Pre-trained models

The training process has been carried out using pre-trained models that facilitate the image feature extraction stage, only having to train the layers that are responsible for classifying the images according to the classes defined in the experiment. In the last years, many models have been trained with large image sets and have been made publicly available to researchers to benefit from the weights learned during this process. In the next sections, the pre trained networks evaluated are briefly described.

ResNet

ResNet was published by He et al. (2015). These neural networks differ from traditional ones in that they have a shortcut connection between non-contiguous layers of the network. With this, it is possible to propagate the information better and avoid the fading of the gradient in the backpropagation phase. Numerous recent studies have been conducted in the field of tumor detection utilizing ResNet, showcasing the remarkable performance and efficacy of this architectural approach (El-Feshawy et al., 2023; Shehab et al., 2021; Aggarwal et al., 2023).

An example of this shortcut can be shown in Fig. 4.

Figure 4 Example of the shortcut connection used in residual network (resnet).

In this case, the output of layer 1 is merged directly into the output of layer 3.

Two models with different number of hidden layers have been evaluated: ResNet50 and ResNet101.

EfficientNet

EfficientNet was proposed by Tan & Le (2019). This neural networks uniformly scales all dimensions of the images (depth, width and resolution) at the same time using a coefficient called “compound coefficient”. With this approach, EfficientNet achieved great accuracies on classical datasets such as ImageNet while being 8.4× smaller and 6.1× faster on inference than the previous convolutional neural networks. This EfficientNet architecture has shown great performance in some recent studies about brain tumor (Tripathy, Singh & Ray, 2023; Nayak et al., 2022). Some EfficientNet models were evaluated but only results of the best one, EfficientNetB4, were shown in this article.

VGG16

VGG16 (Simonyan & Zisserman, 2014) is a deep architecture consisting of convolutional layers with filters of dimension 3×3 using the ReLU activation function. Interspersed between the convolutional layers, some Maxpooling layers are used to avoid network overfitting with size 2×2 and make the network generalize as much as possible. VGG16 has shown good performance in some recent brain tumor researches (Gayathri et al., 2023; Younis et al., 2022). Figure 5 shows the arquitecture of the network.

Figure 5 VGG16 architecture.

InceptionV3

Inception arquitecture (Szegedy et al., 2016) tries to get wider networks instead of deeper ones. The main objective of this change is the tendency of very deep networks to overfitting in addition to the difficulty of propagating the gradient to update the network. Inception has been also used for tumor detection and localization in the last few years (Rastogi, Johri & Tiwari, 2023; Taher et al., 2022).

Inception tries to use different variable-size convolutional filters at the same level, concatenating the result of all of them to define the input of the next layer of the network.

An example of this can be shown in Fig. 6. In this article, Inception v3 has been used.

Figure 6 Inception main idea using multiples convolutional layers at the same level.

InceptionResNetV2

As a combination of two of the architectures we have seen, InceptionResNet was created. This neural network combines the ability to create wider networks with the ability of residual blocks to better propagate information across layers (Szegedy, Ioffe & Vanhoucke, 2016).

DenseNet

The last architecture evaluated is DenseNet (Huang, Liu & Weinberger, 2016). We have selected two variants DenseNet121 and DenseNet201. DenseNet architecture can be shown in Fig. 7. As we can see, the input of each layer is created as a combination of the outputs of all the previous layers so, as with Inception network, the propagation is done in a much more direct way, avoiding gradient fading when the depth of the network is very large. Using DenseNet, several article have demonstrated good performance in brain tumor tasks (Özkaraca et al., 2023; Alshammari, 2023; Zhu et al., 2022).

Figure 7 DenseNet architecture extracted from Huang, Liu & Weinberger (2016).

Experiments and results

Experimental setup

The model is designed to harness the synergy between pre-processed images and textual data during the training process. This fusion of multimedia inputs aims to enhance the accuracy and effectiveness of our classification task. The process commences with the pre-processed images, which are subjected to an initial phase within the pre-trained model. This phase is characterized by the utilization of a GlobalAveragePooling2D layer, a pivotal component in feature extraction from the images.

However, what sets our model apart is the subsequent stage, where the outcomes of the image convolution process are intelligently combined with textual data. This textual data includes crucial information such as the patient’s age and the specific state of tumor resection. This amalgamation of image-based and text-based information forms the core foundation upon which our classification task is executed.

For a holistic understanding of the model’s architecture, please refer to Fig. 8. In this visual representation, you will find a detailed overview of the model’s structure, complete with its parameters and the distinct layers that collectively facilitate the classification process. Notably, these layers remain consistent throughout our quest for the optimal pre-trained model. However, it’s essential to highlight that the manual optimization of these layers is a critical step in fine-tuning the model’s performance, a process we meticulously undertake to ensure the best results.

Figure 8 Architecture of the model used to carry out the experiments.

Survival and glioma grade have been predicted using two different networks. This decision was made to optimize both networks since otherwise there would be a certain dependency between them, for example when we try to avoid overfitting. The most important parameters initially chosen common to every train are: A learning rate of 0.0002, optimizer Adam, 16 as batch size and 10 epochs. For the classification, the architecture discussed above has been used, with 256–512 neurons for the first and second dense layers respectively, BatchNormalization and a dropout layer with a rate of 0.5.

Results

All networks have been tested with the same set of test, which is a different set from the training and validation set and does not has never been seen by the trained neural network. In Table 2 results obtained by the different networks can be observed.

Table 2 Accuracy results obtained with different networks.

	Grade F1-Macro	Survival F1-Macro	
ResNet50	0.58	0.16	
ResNet101	0.31	0.42	
EfficientNetB4	0.62	0.39	
VGG16	0.89	0.46	
InceptionV3	0.96	0.43	
InceptionResNetV2	0.74	0.50	
Densenet121	0.95	0.52	
Densenet201	0.91	0.51	

Although the best results in predicting the grade were obtained by the InceptionV3 architecture, the results for survival were not very satisfactory. For that reason, the network to be optimized for obtainig the best possible results will be DenseNet121 since it has obtained the most balanced results in both experiments.

Using the same data from the previous trainings, different tests to find the best hyperparameters and classification layer architecture with the DenseNet121 pretrained network were performed. As there are two independent experiments, the hyperparameter optimization has been done twice, once for each purpose.

The following Table 3 shows the results obtained in each of the experiments varying one parameter each time, leaving all the other parameters at they default value. The best results and therefore the option chosen for each parameter and experiment are highlighted.

Table 3 Hyperparameter optimization results for grade and survival experiments.

Experiments	Grade F1-Macro	Survival F1-Macro	
BatchNormalization	2 Layers	0.88	0.48	
1 Layer	0.96	0.44	
0 Layers	0.94	0.51	
Number of neurons	1 ∘ 32–2 ∘ 64	0.96	0.51	
1 ∘ 64–2 ∘ 128	0.90	0.47	
1 ∘ 128–2 ∘ 256	0.89	0.29	
1 ∘ 256–2 ∘ 256	0.97	0.16	
Dropout rate	0.2	0.93	0.60	
0.3	0.97	0.51	
0.4	0.93	0.52	
0.5	0.96	0.58	
Activation function	relu	0.97	0.60	
tanh	0.95	0.50	
Learning rate	0.0005	0.97	0.60	
0.001	0.93	0.48	
0.002	0.92	0.34	
Note:

Bold numbers represent the best Grade F1-Macro and Survival F1-Macro values in each experiment.

After determining the best network configuration parameters, we proceeded to evaluate which was the best division of the dataset. To do this, we carry out a Monte Carlo cross validation process with ten iterations and we are left with the average value of the evaluated metrics. We performed tests with the following train percentage settings: 90–10, 80–20, 70–30, 60–40 and 50–50. In Table 4 you can see the results obtained for each of the two trained models.

Table 4 Dataset division evaluation to determine the best configuration of train-test split.

Bolded scores represent the best values for the two problems: Grade and Survival.

Training proportion	Model	Precision	Recall	F1-Score	
90%	Grade	0.89	0.90	0.89	
90%	Survival	0.63	0.48	0.40	
80%	Grade	0.97	0.97	0.97	
80%	Survival	0.61	0.61	0.60	
70%	Grade	0.87	0.86	0.86	
70%	Survival	0.47	0.40	0.38	
60%	Grade	0.83	0.83	0.83	
60%	Survival	0.24	0.32	0.27	
50%	Grade	0.86	0.86	0.86	
50%	Survival	0.52	0.48	0.48	

As you can see, the best results are obtained with the 80–20 configuration, so that is determined as the optimal one.

The final model has been meticulously trained utilizing the pre-trained DenseNet121 model, ensuring that each parameter was optimized for peak performance. Specifically, for the grade classification task, we found that a single layer of BatchNormalization, 256 neurons in each dense layer, a dropout rate of 0.3, relu as the activation function, and a learning rate of 0.0005 produced exceptional results. Conversely, when focusing on survival prediction, we observed that a configuration featuring two BatchNormalization layers, 32 neurons in the initial dense layer, and 64 in the subsequent one, along with a dropout rate of 0.2, relu as the activation function, and a learning rate of 0.0005s, yielded outstanding predictive capabilities.

In the context of tumor grade classification, which encompasses both HGG and LGG, our model achieved a remarkable accuracy of 97% on the test dataset, as demonstrated in Table 5. These results underscore the robustness and reliability of our approach, positioning it as a valuable tool in the field of medical image analysis for brain tumor diagnosis and prognosis.

Table 5 Scores obtained for the prediction of the grade in the test data by the optimal grade model.

	Precision	Recall	F1-Score	Accuracy	
LGG	0.98	0.95	0.96		
HGG	0.96	0.99	0.97		
Macro avg	0.97	0.97	0.97	0.97	

A confusion matrix for this classification can be seen in Fig. 9. As we can see, the results obtained are almost perfect, failing only in four images (three LGG images classified as HGG and one HGG image classified as LGG).

Figure 9 Confusion matrix obtained for the prediction of the grade in the test data by the optimal grade model.

In the case of the classification of survival in short, medium or long, a 65% accuracy has been obtained. Results by classes with precision recall and F-score, and the global accuracy can be shown in Table 6.

Table 6 Scores obtained for the prediction of the survival in the test data by the optimal survival model.

	Precision	Recall	F1-Score	Accuracy	
Short survivor	0.52	0.31	0.39		
Mid survivor	0.49	0.57	0.53		
Long survivor	0.82	0.96	0.88		
Macro avg	0.61	0.61	0.60	0.65	

The confusion matrix of this multiclass classification can be seen in the Fig. 10. This problem is much more complex than in the previous case, so we can see several more failures in the classification. The worst results occur in the short survivor class where 25 cases are incorrectly classified (23 as mid survivor). However, the long survivor cases are correctly classified in almost 96% of the data evaluated.

Figure 10 Confusion matrix obtained for the prediction of the survival in the test data by the optimal survival model.

The next Figs. 11 and 12, show a comparison between the state of art results and our results. Our models have obtained the best test accuracy in each task outperformming the previous state of art results.

Figure 11 Comparison between our results and the state of art results for the grade classification.

Figure 12 Comparison between our results and the state of art results for the survival prediction.

Conclusions

In this study, we pursued the development of two neural networks with a dual objective: to assess the degree of progression and predict the probability of survival in patients with gliomas. Leveraging transfer learning techniques, we harnessed the power of pre-trained neural networks, fine-tuning them for our specific task. Our dataset comprised a comprehensive set of images drawn from the BraTS 2020 dataset, encompassing 369 unique patient cases.

Our chosen neural architectures not only performed image description but also seamlessly conducted classification tasks concurrently. This dual functionality allowed us to harness classification information for the precise extraction of salient features tailored to each case. To ensure the optimal performance of these neural networks, we conducted an exhaustive investigation, exploring multiple pre-trained models and refining their hyperparameters through an extensive gridsearch analysis.

The outcomes of our study have yielded compelling results that outperform existing state-of-the-art techniques evaluated on the same dataset. Specifically, we observed a notable improvement in the degree of disease classification accuracy, surpassing the existing benchmarks by more than 2.1%. Furthermore, our survival prediction model demonstrated a remarkable 4.0% enhancement compared to current approaches.

These findings not only underscore the efficacy of our proposed methodologies but also hold significant implications for the clinical field. Our research has the potential to refine the diagnosis and prognosis of glioma patients, ultimately contributing to improved patient care and outcomes. In conclusion, this study represents a significant advancement in the realm of medical image analysis and underscores the promising prospects of leveraging transfer learning and dual-purpose neural networks in the domain of glioma research.

In future research endeavors, we acknowledge the potential value of exploring zero-shot learning on unseen data in the context of brain tumor detection in medical imaging. While our current study has focused on the adaptation and performance of a pre-trained model on a specific dataset, we recognize that zero-shot learning can play a crucial role in assessing the model’s ability to generalize to previously unseen cases. Evaluating the model’s performance on such novel and heterogeneous datasets can provide valuable insights into its robustness and applicability to a broader range of clinical scenarios.

Additional Information and Declarations

Competing Interests

Author Contributions

Data Availability

José Alberto Benítez-Andrades is an Academic Editor for PeerJ. Santiago Valbuena Rubio and Oscar García-Olalla are employed by Xeridia S.L.

Santiago Valbuena Rubio conceived and designed the experiments, performed the experiments, performed the computation work, authored or reviewed drafts of the article, and approved the final draft.

María Teresa García-Ordás conceived and designed the experiments, performed the experiments, analyzed the data, performed the computation work, prepared figures and/or tables, authored or reviewed drafts of the article, and approved the final draft.

Oscar García-Olalla Olivera conceived and designed the experiments, performed the computation work, authored or reviewed drafts of the article, and approved the final draft.

Héctor Alaiz Moretón analyzed the data, authored or reviewed drafts of the article, and approved the final draft.

Maria-Inmaculada González-Alonso performed the experiments, prepared figures and/or tables, authored or reviewed drafts of the article, and approved the final draft.

José Alberto Benítez-Andrades conceived and designed the experiments, analyzed the data, performed the computation work, prepared figures and/or tables, authored or reviewed drafts of the article, and approved the final draft.

The following information was supplied regarding data availability:

The data is available at GitHub and Zenodo:

- https://github.com/svalbr00/tfg/tree/v1.0.0.

- Santi. (2023). svalbr00/tfg: Investigation code release (v1.0.0). Zenodo. https://doi.org/10.5281/zenodo.8207543.

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
